# Biobanking as a Tool for Genomic Research: From Allele Frequencies to Cross-Ancestry Association Studies

**DOI:** 10.3390/jpm12122040

**Published:** 2022-12-09

**Authors:** Tatyana E. Lazareva, Yury A. Barbitoff, Anton I. Changalidis, Alexander A. Tkachenko, Evgeniia M. Maksiutenko, Yulia A. Nasykhova, Andrey S. Glotov

**Affiliations:** 1Departemnt of Genomic Medicine, D.O. Ott Research Institute of Obstetrics, Gynaecology, and Reproductology, 199034 St. Petersburg, Russia; 2Department of Genetics and Biotechnology, St. Petersburg State University, 199034 St. Petersburg, Russia; 3Faculty of Software Engineering and Computer Systems, ITMO University, 197101 St. Petersburg, Russia

**Keywords:** biobank, genomics, GWAS, meta-analysis, allele frequency

## Abstract

In recent years, great advances have been made in the field of collection, storage, and analysis of biological samples. Large collections of samples, biobanks, have been established in many countries. Biobanks typically collect large amounts of biological samples and associated clinical information; the largest collections include over a million samples. In this review, we summarize the main directions in which biobanks aid medical genetics and genomic research, from providing reference allele frequency information to allowing large-scale cross-ancestry meta-analyses. The largest biobanks greatly vary in the size of the collection, and the amount of available phenotype and genotype data. Nevertheless, all of them are extensively used in genomics, providing a rich resource for genome-wide association analysis, genetic epidemiology, and statistical research into the structure, function, and evolution of the human genome. Recently, multiple research efforts were based on trans-biobank data integration, which increases sample size and allows for the identification of robust genetic associations. We provide prominent examples of such data integration and discuss important caveats which have to be taken into account in trans-biobank research.

## 1. Introduction

The term “biobank” is defined as a structured collection of biological samples and associated data, stored for the purposes of present and future research [1]. The first such collections of biological samples were established long before the beginning of the genomic era; however, large population-based collections were mostly used in epidemiology rather than genetics [2]. The advent of multiple omics technologies, mostly based on next-generation sequencing (NGS) methods [3], substantially increased the role of biobanks in research. Given this development, modern biobanks are involved in a wide variety of efforts that range from biosample collection and clinical/lifestyle data storage to complex trans-biobank collaborative studies. In our review, we summarize the main directions in which biobanks aid genomic research, and provide successful examples of biobank data usage in recent scientific endeavors. We also extensively describe the aims of establishing the largest biobanks, their capacity, sample storage, ways of collecting clinical/lifestyle data, and potential for genomic studies. Finally, we describe major research efforts involving trans-biobank data integration, and discuss future prospects for such studies.

## 2. Applications of Biobanking in Genomic Medicine and Research

Many recent developments in the field of human genetics and genomics would not have been possible without biobanks. There are many directions in which biobanks benefit modern human genomics. Such main applications of biobanks in medical genetics and genomics are summarized in Figure 1.

First, biobank-scale genetic datasets are an excellent source of population frequencies of genetic variants. Collection of such allele frequency (AF) information was one of the first motivations for the establishment of genetic data collections and biobanks [2]. Several successful attempts to construct a “population reference genome” or local AF databases were performed recently in UAE [4], Japan [5], the Netherlands [6], and Russia [7,8].

Efforts to construct such AF references are motivated by the presence of many population-specific genetic variants that are not represented in the reference human genome or global allele frequency databases such as the Genome Aggregation Database [9]. Population allele frequencies are especially relevant for clinical geneticists who are interpreting the results of NGS-based tests in molecular diagnostics of inherited disorders. For example, filtering candidate genetic variants by population maximum (popmax) AF rather than global AF significantly decreases the burden of variant interpretation [10]. Moreover, allele frequency information is required for the clinical classification of genetic variants in accordance with the widely used American College for Medical Genetics and Genomics (ACMG) guidelines for variant interpretation [11].

Another application of allele frequency information is the analysis of the genetic structure of human populations. For example, AF data can be used to study migrations and admixture of ancestral populations (e.g., in a milestone work on the population of modern India by [12]). Moreover, comprehensive AF resources facilitate the analysis of complex patterns of evolutionary conservation in the human genome and identification of clinically significant genes and genome regions [13,14].

Second, whole-genome sequencing (WGS) data collected by biobanks can be used for the construction of the local reference genotype panels. Such panels provide important additional information about genetic variation. Examples of biobank-based reference panels include the SISu reference panel for the Finnish population (http://sisuproject.fi, accessed on 6 June 2022) or a recently developed WGS-based panel for the United Kingdom [15]. Reference genotype panels can be used for genotype imputation, i.e., prediction of an individual genotype at genome locations not covered by the genotyping array. Imputation increases the power of genome-wide association studies (GWAS) and facilitates meta-analysis [16]. Furthermore, reference genotype panels enable statistical fine-mapping, i.e., identification of the actual causal genetic variants at GWAS loci (reviewed in [17]). Finally, information on linkage disequilibrium (LD) between genetic variants and reference haplotypes for a population can be important for multiple research purposes, such as the analysis of recent positive natural selection (reviewed in [18]) or in-depth analysis of the population structure. For example, a recent analysis of a large set of genomes that revealed interesting novel details about the British population, including region-specific rates of relatedness [15]. Another application of the reference haplotype information is genotype phasing, which is important, in particular, in rare disease diagnostics [19].

Third, samples collected by biobanks can be used by researchers in various types of high-throughput assays. Genotyped samples can be used as external controls in GWAS for rare or common diseases. Integration of external controls in case-control studies has been shown to improve the power of rare variant association [20]. Such external controls are frequently drawn from open genotype datasets such as the UK Biobank dataset (discussed below). Apart from the direct inclusion of external controls in the analysis, reference data generated by biobanks might be used in other types of analysis. For example, genotype and phenotype data can be used to define exposures for Mendelian randomization (MR), a group of methods designed for testing a causal relationship between traits (reviewed in [21]).

Fourth, information collected by the biobanks, including comprehensive questionnaires, physical measurements, health check-ups, and data from hospital databases and national health registries, makes an extensive data array for association studies. Numerous single-trait GWAS have been performed using biobanks, and such studies greatly enhance our understanding of human genetics and disease biology (reviewed in [22]). The results of these studies enable the development and evaluation of polygenic risk scores (PRS) (reviewed in [23]). Implementation of PRS into clinical practice for the prediction of individual risk is a cornerstone of present-day personalized medicine [24]. Population-level biobanks may enable the development of embryonic polygenic risk scoring for preimplantation genetic testing and implementation of such risk scores in clinical practice [25].

However, the real power of biobanking lies not in single-trait analyses, but rather in the ability to perform comprehensive analyses using multiple traits in the phenotype. For several biobanks, such as the UK Biobank or the FinnGen project, GWAS results for thousands of phenotypes are made publicly available to the scientific community. Aggregation of such publicly available data allows for phenome-wide association studies (PheWAS), as well as identification of genes that affect several traits or diseases in the phenotype (such genes are termed pleiotropic) (e.g., [26,27]).

Finally, biobanking facilitates the movement of healthcare facilities toward precision medicine. Genome-informed approaches are being established by returning genomic results to participants of cohort studies. As shall be outlined below, such efforts are made by the Tohoku Medical Megabank Organization [28], Estonian Biobank [29], and All Of Us Research Program [30]. There are many concerns about which genes should be examined without a specific diagnostic purpose, and which results are to be returned. The results should be clinically significant, which implies that the functional impact of the variant and its connection to the disease have to be carefully ascertained. Furthermore, the healthcare system should have protocols for managing patients with diseases associated with selected genes.

As mentioned above, germline variant pathogenicity is determined according to the ACMG guidelines for the interpretation of sequence variants [11]. The somatic variants are classified by the International Agency for Research on Cancer (IARC) [31]. Among five groups of variants the most arguable in clinical practice are variants of uncertain significance (VUS). The guidelines of EuroGentest and the Canadian College of Medical Geneticists suggest the necessity of periodic reanalysis of VUS, but do not advise on whether these variants should be reported or not [32]. According to the ACMG, VUS should not be considered in the clinical decision-making process [11].

Another concern of analyzing genomic data is unsolicited (or incidental) findings (UFs)—variants that are out of the scope of the initial purpose of genetic testing, but medically significant for the patient or family member(s) [33]. There is still a lack of consensus about reporting UFs. European Society of Human Genetics recommends limiting genomic analysis “as targeted as possible”, and recent updates suggest the necessity of overall evaluation of exploring UFs before introducing it in general practice. The Genomics of England and ACMG developed an approach of reporting findings on the basis of informed consent in a deliberate clinically significant set of genes, which do not correspond to the primary purpose of genomic test [34]. The ACMG published the guidelines devoted to managing such secondary findings and periodically updates the list of medical valuable genes in which all disocvered variants with sufficient support for pathogenicity should be reported. The genes were prioritized by the high penetrance and presence of effective preventative measures on the asymptomatic stage of associated disease [35]. Despite the fact that these recommendations are devoted to clinical genetic testing, the mentioned criteria could be assumed in the policy of returning genomic results to participants by biobanks as well. Moreover, ethical, psychological, and security issues should also be considered when designing such policies.

## 3. World’s Largest Biobanks and Genomic Research

Due to the rapid development of biobanking, economically developed countries possess at least 11 biobanks per million population [36]. Such a number makes it impossible to review all projects and provide an extensive list of studies that are based on biobank data.

Hence, in the following sections we will focus on the largest and most well-known biobanks and multi-biobank projects that not only collect samples and abundant phenotypic information, but also participate in large-scale multi-biobank research initiatives (see below). We will describe in more detail, the biobanks that have been extensively used in genomic research or have a future prospect of becoming a rich genomic resource (Table 1).

### 3.1. UK Biobank (UKB)

UK Biobank is arguably the most widely known and actively used biobank in the field of human genomic research. For example, the term “UK Biobank” is mentioned approximately 47,200 times according to Google Scholar Citations, with the closest competitors having less than 5000 references. UKB is a large prospective population-based database, which explores the occurrence and development of diseases of middle and old age [75]. Between 2006 and 2010, primary data from 500,000 UK residents aged 40–69 years were collected [76]. Baseline questionnaires were designed to accommodate cutting-edge research questions. During the interview, hearing, and cognitive functions were assessed. In research centers participants underwent anthropometric measurements, hand grip strength test, spirometry, and densitometry. Additionally, blood and urine were tested for biochemical parameters. Levels of HbA1c and rheumatoid factors were measured [77]. Blood, urine, and saliva samples were preserved. Follow-up observations are designed for 20 years and include data collection from medical and demographic registries. (https://www.ukbiobank.ac.uk/, accessed on 12 April 2022).

UKB boasts a rich collection of genomic data, including genotyping array, whole-exome (WES), and WGS sequencing. There are multiple examples of how these datasets were applied for both singlet-trait and phenome-wide analyses. In 2020, UKB released a large dataset of 49,960 WES samples that was used to discover previously unknown associations between pLoF variants in specific genes and different traits, including *PIEZO1* for varicose veins and *MEPE* for Becker muscular dystrophy [39]. Currently, WES and WGS data are already available for nearly 300,000 and 200,000 participants, respectively. Analysis of 150,119 whole genomes identified numerous novel SNPs and structural variants, and revealed notable associations between rare structural variants with various traits (e.g., a deletion in *PCSK9* was found to be associated with non-HDL cholesterol levels) [15].

Genotyping array data generated by UKB have been extensively used for GWAS. In its 2018 study, UKB performed GWAS using approximately 850,000 variants in all participants [37]. New findings from these data are too numerous to cite; one example is a recent identification of genetic predictors of obesity complications: steatosis, cirrhosis, hepatocellular carcinoma [40]. Findings include the rs738409 in *PNPLA3* and a male-specific rs58542926 variant in *TM6SF2*. A notable feature of UKB data is the inclusion of multiple self-reported traits. For example, such phenotypic information was used in a recent GWAS for hearing impairment [38]. In this analysis, 44 associated loci reveal the multifactorial nature of the disorder based on the involvement of these genes in the wide spectrum of biological functions. A remarkable amount of 34 genes identified in this study were novel risk loci.

### 3.2. BioBank Japan (BBJ)

BBJ was the first patient-based biobank, which initially focused on 47 common diseases. From 2003 to 2008, it recruited approximately 200,000 participants with newly developed diseases and ones who were diagnosed and treated before the project began [78]. Additionally, in 2013-2017 60,000 participants with at least 1 of 38 chosen diseases were enrolled (https://biobankjp.org/en/info/nbdc.html, accessed on 5 June 2022). In a baseline survey, BBJ collected information about lifestyle, medical history, and results of physical measurements. DNA, serum, and tumor tissues were collected and stored. Follow-up data were obtained by analyzing medical annual records, and vital statistics were requested from hospital and national registries [78].

BBJ provides large-scale genomic data for discovering genes and variants’ associations with common diseases. SNPs data are available for approximately 180,000 BBJ participants. WGS has already been performed for 3,000 DNA samples, metabolome/proteome analyses have also been conducted (https://biobankjp.org/en/info/nbdc.html, accessed on 5 June 2022). Massive GWAS of 42 diseases revealed several novel loci [41]. As an example, the identified association of missense variants in *ATG16L2* and coronary artery disease had not been identified in previous studies of European populations. In another interesting study, GWAS of food and beverage preferences uncovered *ALDH1B1* and *ALDH1A1* genes linked to enzymes of alcohol metabolism [42]. Missense rs8187929 in the *ALDH1A1* had not been previously found in European ancestry individuals. These examples illustrate the power of biobank-driven discovery of population-specific risk factors for common disease.

Some genomic and clinical data from BBJ is available on the website of the National Bioscience Database Center (NBDC). Merged WGS data from the disease-oriented BBJ and the population-based ToMMo cohort resulted in GEM-J WGA—a publically accessible—variant variant frequency panel for the Japanese population (https://togovar.biosciencedbc.jp, accessed on 5 June 2022).

### 3.3. FinnGen Research Project

FinnGen is a collaborative project of universities, medical clinics, and biobanks in Finland with the aim of finding genome-disease associations. The significance of the project for the world community is explained by the relative genetic isolation of the Finnish population and the high frequency of many initially rare variants [79]. In 2017, FinnGen launched a study of 500,000 participants. 200,000 samples were chosen from previous cohort studies devoted to various disorders, and another 300,000 biosamples have been collected in medical centers or have been taken from healthy volunteers (https://www.finngen.fi/en, accessed on 10 April 2022). Participants’ data are received from the national health and hospital databases [44].

To control the quality of obtained phenotypic data, FinnGen used 15 diseases with over 1000 cases in the FinnGen cohort and previously published GWAS results [44]. In addition to confirming the validity of the data, 30 new associations were found with prevalence among the population of Finns and neighboring Uralic language families. WGS results from more than 3500 participants revealed 16,962,023 population-specific SNPs and indels which were imputed in the genome-wide association analysis. GWAS results for more than 1900 clinical conditions defined in medical registries have shown the possibility of detecting associations for variants with low frequency. For example, an association between a missense variant in *STAB2* and venous thromboembolism confirmed recent studies of this gene as a potential biomarker [43].

FinnGen has collected the genotype-phenotype data of 392,000 participants. After a one year embargo, summary statistics of GWAS are published in open access on the project website. Researchers from partner organizations are allowed to access up-to-date information.

### 3.4. Estonian Biobank (EB)

Estonian Biobank was founded in 1996 to advance medical care through genetic studies [80]. Nowadays, a cohort of over 200,000 Estonian citizens aged 18 or over represents the general adult population of the country.

The questionnaire covered topics related to lifestyle, educational background, occupation, and personal and family health history. A whole blood sample was used for biochemical profiling, DNA, white blood cells, and plasma isolation, while a buccal swab was collected for RNA extraction. GWAS results are available for 200,000 samples, as well as WGS and WES data for more than 2500 samples (https://www.eithealth-scandinavia.eu, accessed on 15 March 2022). Numerous transcriptomic, metagenomic, and metabolomic studies have already been conducted using EB samples. In addition, 2700 blood samples were examined for 42 biochemical tests, telomere length was measured in 5200 participants. It should be pointed out that the government legislated the Human Genes Research Act (HGRA), which regulates EB activity. Together with the broad informed consent, it allows obtaining participants’ information on health status, prescribed medicines, and causes of death from national registries and hospital databases and implementing it in a broad spectrum of studies without re-consent [81]. At the same time, HGRA stated that participants are entitled to receive their personal data stored in the biobank (except for genealogies), or decline to receive any such information [82]. These measures certainly support advances in scientific research and the application of its findings to clinical medicine.

EB provides notable examples of how genetic test results from cohort studies can be utilized in clinical practice. In a 2019 study, the EB team reported a systematic recall and (upon receiving a positive response from the participant) re-examination of participants and their family members with mutations in *LDLR*, *APOB*, *PCSK9* genes [29]. These pathogenic variants in these genes are related to familial hypercholesterolemia (FH), which causes a 20-fold increase in the risk of early-onset coronary artery disease compared to the general population. Among identified carriers of screened FH mutations, only half had already been diagnosed with nonspecific hypercholesterolemia and took statins. In another notable study, genotype data obtained through different NGS-based methods in over 44,000 participants of EB were used to choose optimal methods of discovery, validation, and accounting variants in genes involved in drug response [46].

### 3.5. China Kadoorie Biobank (CKB)

CKB is a collaborative cohort study initiated on account of the increased role of chronic diseases in death and disability among Chinese people [83]. Data about the main collection of 512,891 individuals aged 30–79 from 5 urban and 5 rural regions of China were obtained between 2004 and 2008 [84]. The baseline data array included anthropometric measurements and questionnaires related to general demographic and socio-economic topics, lifestyle habits, medical history, physical activity, and reproductive history for women. Blood samples were taken and tested for glucose level and HBs-Ag. The estimated duration of observations is at least 20 years. Repeated surveys are conducted among randomly selected 5% respondents [84]. Morbidity, hospitalization, and mortality data are available from national registries.

CKB has generated large-scale genotyping data for more than 100,000 participants. Moreover, WGS of the entire cohort, metabolite, and DNA methylation array results are underway [85]. Available results have already allowed large-scale GWAS for populations of non-European descent, which are crucial for genetic heterogeneity evaluation. For example, GWAS of the lung function parameters in the Chinese population identified novel loci for forced vital capacity (FVC), forced respiratory volume in 1 second (FEV1) and their ratio (FEV1/FVC) [49]. A higher risk of development of respiratory diseases and reduction of lung function among subjects with obesity led researchers to discover a shared genetic component between lung function and obesity in the Chinese and European populations. CKB participates in large-scale studies of common diseases such as type 2 diabetes [47] and depression disorders [48] in East Asians. Another interesting example is a recent analysis of the genetics of fingerprint patterns [50].

### 3.6. Tohoku Medical Megabank Project (TMM)

Tohoku Medical Megabank was established in 2011 to improve the healthcare system of the prefectures affected by the Great Japan Earthquake [86]. As thus, the study represents an important example of how biobanking can be useful for research related to large-scale crises and natural disasters.

Recruitment into the population-based TMM CommCohort, and the TMM BirThree cohort began in 2013 [87]. In the TMM CommCohort study, about 84,000 residents of Miyage and Iwate prefectures aged 20–74 were asked to complete a baseline questionnaire, supplemented with an inquiry about disaster impact. Participants underwent eye examination, dental checkup, hearing test, spirometry, densitometry, and hand grip strength test. Blood and urine tests were performed. Some respondents provided MRI data. Follow-ups consist of periodic questionnaires and collection of information from vital databases and medical registries [88]. The TMM BirThree Cohort Study has recruited about 73,000 pregnant women and their family members, with the aim to study common diseases in disaster-affected areas (e.g., infectious, allergic diseases, developmental disorders). TMM stores blood samples, urine, saliva, breast milk, dental plaque, stimulated T-cells, and EBV-transformed B-cells (https://www.megabank.tohoku.ac.jp/, accessed on 2 April 2022).

TMM established the jMorp database to integrate proteome, metabolome, genome, methylome, and transcriptome data from the cohort participants. The database includes population frequencies of SNPs from 14,000 WGS samples [53]. Using the jMorp Multi-Omics Panel, the total carrier frequency of Pompe disease in Japanese has been evaluated [55]. The jMorp AF reference also enhanced filtering of false positive findings of NGS-based tests, imrpoving the diagnosis of rare diseases in Japan [52].

The results are also being used in pharmacogenomics. For example, seven novel SNPs in *CYP2B6* were identified in 1070 TMM participants. The variants were tested for their effect on the enzymatic activity of the gene product, cytochrome P450 2B6, which participates in the metabolism of common drugs (e.g., bupropion, efavirenz, cyclophosphamide, ketamine) [51,89]. The information on the patient’s genotype and the corresponding variants’ effect on drug metabolism could influence drug selection, minimize adverse effects and maximize therapeutic efficacy.

A series of pilot studies of returning individual genetic results to patients was launched by the TMM project. Two studies on relatively small cohorts of TMM participants, who met the inclusion criteria and confirmed their agreement by signing informed consent, have been published recently. The first study focused on familial hypercholesterinemia (FH)—monogenic and high-penetrant disorder [28]. The second pilot experiment was devoted to implementing pharmacogenomics results into clinical practice. Persons, who were identified as carriers of polymorphisms in the chosen genes—*MT-RNR1*, *CYP2C19*, or *NUDT15*—received recommendations both for themselves and healthcare workers [54].

### 3.7. Taiwan Biobank (TWB)

One more prominent East Asian biobank was established in Taiwan in 2012. The prospective community-based cohort represents the population structure and involves citizens 20–70 years old with no prior diagnosis of cancer, among whom 99% belong to Han Chinese ancestry. The baseline data were collected through questionnaires with culture-specific sections, anthropometric measurements, medical imaging (abdominal ultrasound, bone densitometry, ECG), as well as collection and analysis of blood and urine samples. TWB expands data collection by active follow-ups every 2–4 years and future integration to National Health Insurance Research Database and over 70 health registries. At present, the information from the aforementioned sources could be accessed only upon demand after an institutional review board approval. WGS, DNA methylation, metabolomics, and HLA typing data are available for subsets of the community cohort [90].

To improve the imputation efficiency, a population-specific TWB reference panel was generated based on the WGS data from 1445 cohort participants. Two custom genotype arrays were designed on which about 103,000 samples have been genotyped [56]. A recent study revealed the *NOTCH3* p.R544C variant as a valuable risk factor of ischemic stroke in Taiwan population [91]. In another study, PheWAS of 10 diseases and 34 quantitative traits identified more than 900 significantly associated loci among which 100 loci are population-specific [57].

TWB shares genomic summary results on Taiwan View website (https://taiwanview.twbiobank.org.tw/index, accessed on 13 November 2022). Depersonalized individual-level data could be obtained upon request, except information about Taiwanese indigenous peoples. (https://www.twbiobank.org.tw, accessed on 13 November 2022). In addition, TWB has been working on the legislation of returning UFs to the cohort participants [92].

### 3.8. LifeLines Cohort Study

LifeLines is a multigenerational cohort study of the inhabitants of the northern part of the Netherlands, which is aimed to study the issues of healthy aging and the etiology of chronic diseases that are relevant to the European population [93]. The prospective cohort parameters represent the population of the region [94]. People without severe psychiatric or physical illness were recruited and asked to attract family members to the study. As a result, 51% of study participants were part of a two-generation relationship and 12% of a three-generation relationship.

Phenotypic and environmental data were obtained through age-adapted questionnaires, anthropometric measurements, laboratory sample tests, spirometry, electrocardiography, cognitive tests, and skin autofluorescence [93]. Cohort participants received the results of a medical examination. Follow-ups are estimated for at least 30 years. The database is extended by data from administrative health databases (cancer registries, vital statistics, and health insurance databases). Every one and a half years participants fill out a questionnaire and every 5 years are invited to a health check-up.

Among notable projects run by LifeLines, it is important to mention the Digestive Health project that focuses on the gut microbiome, metabolome, and transcriptome studies. The collected data have already allowed scientists to analyze the effects of human genome-gut microbiome interaction on the regulation of blood proteins associated with cardiovascular diseases [61]. Numerous associations of variants, diet patterns, and gut microbiota composition have been identified [59]. Proton pump inhibitors have been shown to reduce the diversity of the gut microbiota more than other common drugs, increasing the risk of intestinal infections [60].

### 3.9. Other Biobanks

As mentioned at the beginning of the section, the total number of biobanks makes it difficult to provide an exhaustive list of these projects and their application examples. However, several other projects deserve a brief discussion.

Karolinska Biobank was established in 2004 at Karolinska Institute [95]. The biobank includes project-specific cohorts, with LifeGene and TwinGene studies being the most remarkable examples. The TwinGene cohort comprises phenotypic data and biological samples of twins born before 1958 ([96], https://ki.se/en/research/biobanks-and-registrie, accessed on 2 June 2022). This cohort is a very promising resource as twins have long been used as the main model in human genetic epidemiology [97]. The TwinGene cohort was used in a recent GWAS meta-analysis which identified many new risk loci for irritable bowel syndrome [64,98].

National Biobank of Korea (NBK) is a cross-country biobank platform that manages population-based cohorts (Korean Genome and Epidemiology Study (KoGES), the Korea National Health and Nutrition Examination Survey) as well as regional hospital-based collections, which store samples and associate data from 439,602 and 612,185 participants respectively. NBK and 2 collaborative biobanks are parts of the Korean Biobank Network (https://nih.go.kr/biobank/cmm/main/engMainPage.do, accessed on 16 November 2022), which was established to provide security to biobanks’ data and implement unified guidelines for biobanks’ procedures ranging from sample collection to biospecimen distribution. Based on large-scale NBK data, a genotyping array was designed and optimized for the identification of trait-associated variants specifically in the Korean population [63]. KoGES data contributed to a better understanding of the genetic architecture of electrocardiographic characteristics [99], T2D [100], and several other traits in Asians [101].

Among other large biobanks, the HUNT Biobank in Norway and the Canadian Partnership for Tomorrow’s Health (CanPath) are also worth mentioning. The HUNT Biobank was established in Nord-Trøndelag, and four separate studies (HUNT1–HUNT4) were being conducted since 1984 [102]. In the most recent iteration, samples of saliva and feces were collected for microbiome studies [103]. The HUNT cohorts were used for GWAS for atrial fibrillation [65], impaired mineral metabolism [67], and LDL levels [66]. CanPath also possesses a rich set of data which was used for association analyses of kidney dysfunction [69] and other traits such as hair color [68]. In this study, loci associated with hair color also affected skin cancer development.

Several large biobanks store large amounts of phenotypic and environmental data, but are not used for public genomic research. Biobank Graz stores a wide variety of biosamples (e.g., whole blood, CSF, tissue samples) and data for more than 1,200,000 patients of the departments of University Hospital Graz (https://biobank.medunigraz.at/en/who-we-are, accessed on 6 April 2022). The collection consists of over 20 disease-specific cohorts as well as a healthy control group [104]. Among others, Shanghai Suburban Adult Cohort, and Biobank are unique resources for studying a population response to sudden changes in lifestyle, ecology, and economy due to worldwide urbanization [105]. This resource, however, has not yet been applied for genomic analyses.

Another noteworthy study was initiated by the National Institutes of Health (NIH) to implement precision medicine into the health system of the USA. The particular concern of the All of Us Research Program is demographic groups that have been and remain underrepresented in biomedical research [30]. The program is expected to recruit at least 1 million people into the cohort [106]. In March 2022, the first genomic dataset with 98,000 WGS and nearly 165,000 genotyping array results were released (https://www.nih.gov/, accessed on 2 April 2022).

Several initiatives were run in Russia, where numerous biobanks were created or formed from existing collections of human and environmental biospecimens. In December 2018 biobanks were united into the Russian National Association of Biobanks and Biobanking Specialists (NASBIO) [107]. A joint genetic project of the Biobank of the National Medical Research Center for Preventive Medicine (Moscow, Russia) and the Biobank of Saint-Petersburg State University provided a major boost to the project. The joint study was dedicated to the discovery of novel gene targets associated with obesity and type 2 diabetes. Several highly case-specific variants in genes previously not directly linked to type 2 diabetes and/or obesity (e.g., *TMC8*, *PCDHA1*, *PLEKHA5*, *HBQ1*, *VAV3*, and *ADAMTS13*) were detected [108].

## 4. Data Integration and Prospects of Trans-Biobank Research

The range of aforementioned benefits of biobanking in various populations in both research and clinical applications can be further expanded by considering the ever-growing number of trans-biobank collaborative research. A steady increase of interest in the creation of biobanks, especially in previously underrepresented non-European descent populations, provides ample opportunities for meta-biobank studies that could shed the light on the genetics of diseases, improve our understanding of disease classification and interconnection between clinical conditions and widen the translation of research findings into clinical practice.

The movement to consolidate biological sample collection already has many prominent examples [107,109,110,111]. Aims of such associations are manifold, including consolidation of biological material to increase its availability to researchers, standardization of research practices and training of personnel for biobanks, production, and publication of associated data to the scientific community, and granting access to these data to scientific users. Networks of biobanks provide immediate benefits of improved functioning resulting from better organization of data and harmonization of standard operating procedures. Another advantage in the field of biobanks’ interaction lies in the creation of a shared infrastructure that includes information systems that address the issue of availability and consistency of the biobank data [112]. But perhaps most important for genomic research is the connection of genomic data with a vast compendium of phenotype data coming from electronic medical records, questionnaires, and other sources. This type of trans-biobank catalog empowers association studies especially when large cohorts of participants are hard to collect.

Many of such trans-biobank research projects focus on particular conditions and their genetic determinants across various ancestries. Examples include thoracic aortic disease [113], type 2 diabetes [114], prostate cancer [115], neurodegenerative diseases [116]. The main mode of usage of genomic data across biobanks in these studies is the meta-analysis of genome associations found across populations. In most cases, such meta-analysis is used to replicate associations observed in the discovery cohort. However, the replication rate is frequently far from perfect. As an example, a recent cross-trait meta-analysis of CKB and UKB data identified a common relationship between lung function and obesity in the two populations. At the same time, the exact loci associated with these traits in two populations had a very low degree of overlap [49].

One of the most prominent and global meta-analysis efforts belongs to the COVID-19 Host Genetics (COVID-19 HG) consortium [117]. The project currently comprises 119 registered studies (https://www.covid19hg.org/partners/, accessed on 4 July 2022). A meta-analysis included 49,562 patients from 46 studies identified 13 loci associated with COVID-19 infection or disease severity, including such genes as *ABO*, *SLC6A20*, and others. Associations in *ABO* and *SLC6A20* were later replicated in another association analysis including multiple large-scale studies [118]. Furthermore, a gene-level trans-ancestry meta-analysis of whole-exome sequencing data from 21 cohorts identified *MARK1* and *TLR7* as novel COVID-19 risk genes [119].

However, research efforts involving trans-biobank data aggregation are not limited to the analysis of individual traits or diseases. For example, a recent study by Sun et al., integrated WES data from UKB and genotyping array data from FinnGen to identify causal coding variants for 744 disease endpoints [45]. This integrated analysis revealed 975 associations; moreover, results demonstrated a role in complex diseases for variants previously linked to Mendelian disorders.

A notable example of a study involving more than two datasets comes from a cross-population study of 220 phenotypes matched between UK Biobank (UKB), FinnGen, and Biobank of Japan (BBJ). The authors studied several parameters of correspondence between the biobanks. From the standpoint of pleiotropy, BBJ, and European biobanks had similar loci associated with the largest number of traits even corrected for correlated genotypes or closely associated phenotypes. Most notably the MHC locus scored highest in European populations while in the Japanese cohort it was preceded by the *ALDH2* locus. Across all studied biobanks, pleiotropic loci were associated with recent positive selection as measured by singleton density score, highlighting shared patterns of natural selection in different human populations. Analysis of genetic correlations also proved that many conditions shared genetic etiology across human populations, despite differences in populations themselves and medical and diagnostic practices.

Another benefit from the combination of biobanks comes from the ability to elucidate and explain common genetic architecture of diseases by application of matrix decomposition of disease GWAS summaries and characterization of resulting hidden components. This approach may also be applicable to sub-significant associations which are instrumental to understanding the contribution of common variants to rare diseases and when studying underrepresented populations. In the comparison study of BBJ, UKB, and FinnGen components resulting from the matrix decomposition explained converging etiology of multiple similar conditions, allowed to interpret an underpowered GWAS of varicose vein in BBJ by matching it with more powered GWAS in UKB and provided an approach to categorize diseases based on matrix decomposition components. The authors additionally complemented components with results from metabolome GWAS and biomarker GWAS underscoring the importance of trans-biobank studies, including a combination of biobanks of different modalities [120].

Integrated data of BBJ, FinnGen, and UKB have also been used for the construction of PRS for complex traits that could help identify the driver biomarkers affecting human lifespan [121]. That way genetic data could be translated to the clinic as these complex factors could be modified through medical treatment. High PRSs of several biomarkers were found to be implicated with lifespan across different ethnicities. For example, high systolic blood pressure (sBP) was significantly associated with a shorter lifespan in all studied populations, and among cause-specific mortality factors, cardiovascular and cerebrovascular diseases were associated with sBP. At the same time, hypertension was not significantly interacting with lifestyle factors, suggesting the high efficacy of lifestyle interventions even for individuals with high genetic risks. Interestingly, associations of body mass index (BMI) PRSs with lifespan were different between European (UKB and FinnGen) and Japanese populations which could be explained by the ethnic differences in the health burdens of obesity between Japanese and European individuals, underscoring the importance of trans-ethnic studies and warranting further investigations in the field.

Another interesting example of novel insights obtained from cross-biobank data integration is the recent identification of sex-specific participation biases in biobanks [122]. In this study, genotype, and phenotype data were integrated from UKB, FinnGen, BBJ, and other projects to analyze genetic variants associated with the participant’s sex. Remarkably, the authors identified numerous loci showing significant association with sex. Further analysis of this unexpected result showed that sex differences were substantial only in projects with active recruitment of participants (23andMe and UKB). Moreover, sex-differential participation in UKB was linked to educational attainment levels in participants of the opposite sex.

The largest to date effort to bring together findings from multiple biobanks is represented by Global Biobank Meta-analysis Initiative (GBMI) [49]. This collaborative network unites 24 biobanks across four continents representing more than 2.2 million individuals with both genomic and phenotypic data from six main ancestry groups: approximately 33,000 of African ancestry either from Africa or from the admixed-ancestry diaspora (AFR), 18,000 admixed American (AMR), 31,000 Central and South Asian (CSA), 341,000 East Asian (EAS), 1.4 million European (EUR), and 1600 Middle Eastern (MID) individuals. This type of cooperative effort utilizing summary statistics across biobanks makes possible several types of scientific goals including increasing the power of GWASs for common diseases, enabling the genetic investigation into less prevalent or understudied diseases, increasing the ancestral diversity of genetic association studies, and in doing so analyzing a broader set of genetic variation, cross-validating new findings across biobanks. The authors conducted a meta-analysis of GWASs on 14 diseases and endpoints, including rare and sex-specific diseases across a 30x prevalence range. Across all biobanks, more than 70 million variants were tested for associations including 2.9 million protein-coding variants.

Inverse variance-based meta-analysis replicated previously reported findings and identified apparently novel associations of which nearly half are either protein-coding or are in linkage disequilibrium with coding variants. The authors additionally demonstrated the benefit of adding non-European ancestry samples. Nine out of 499 loci used for effect size comparison across ancestries showed evidence for heterogeneity of effect sizes. The difference in effect sizes was also observed in sex-stratified analyses.

In general, GBMI is composed of several scientific groups working on elucidating the genetic architecture of phenotypic endpoints and its biological implications and characterization of discovered associations via fine-mapping of loci, studying transcriptome-wide associations, prioritizing drug targets, and improving disease risk prediction.

There are several important concerns regarding data integration between large-scale projects. First, population structure is an important factor that may influence the results of cross-ancestry data integration (reviewed in [123]). Genetic differences between populations may affect the results of statistical analyses that are reliant on haplotype structure (e.g., heritability or genetic correlation estimation by LD-score regression [124]). Another method that may be biased by population structure is the two-sample MR which is extensively used to evaluate causal relationships between traits. For example, a recent study of intracranial aneurysms conducted jointly by multiple groups, including the China Kadoori Biobank and BioBank Japan [125], discovered a causal relationship between blood pressure and aneurysms. The latter analysis was done using two-sample MR with UK Biobank data used as reference exposures. While the results of the analysis corroborate earlier clinical findings, two-sample MR may yield inadequate results in such studies.

Second, phenotyping procedures and encoding used by different biobanks may vary. The definition of phenotypes itself is a difficult issue, and different standards have been proposed for this purpose [126]. Many studies use the International Classification of Diseases (ICD-10) codes. However, some studies use modified versions of ICD-10 (e.g., FinnGen [44]), making it harder to match traits across biobanks. Usage of the Human Phenotype Ontology (HPO) terms [127] also seems a promising strategy. Today, HPO is more frequently used in the field of rare disease diagnostics. At the same time, phenotype data provided by the biobanks have to be additionally converted to HPO terms (such a conversion has been applied to UKB data, e.g., [128]). An important advantage of HPO is that it attempts not only to standardize the description of traits, but also to provide an accurate classification that allows researchers to use different levels of granularity in phenotype description and/or data analysis (e.g., when analyzing pleiotropy [129] or relationships between genes and traits [130]). Hence, as the number of trans-biobank research efforts grows, it is likely that HPO (or other standardized phenotype description resources) will be used more extensively, facilitating various types of research. The problem seems to be especially relevant for predictive model construction and artificial intelligence (AI) driven discovery.

Third, data preprocessing and statistical tools used for analysis may also result in biases specific to a particular dataset. For example, the integration of publicly available GWAS summary statistics from UKB and FinnGen requires two important steps. First, UKB genotypes are given with respect to the GRCh37 human reference genome assembly, while FinnGen uses GRCh38. Hence, one of the datasets has to be lifted over to another assembly prior to any analysis. Second, variations in phenotype pre-processing pipelines may result in different effect size values for the same trait (Figure 2a). These differences may in turn affect the results of meta-analysis, at least in some widely used methods such as the inverse variance-based test in METAL (Figure 2b, left) [131]. Standardization or scaling of effect size values could compensate for this issue (Figure 2b, right); however, the validity of such scaling may require rigorous proof.

In addition, it should be considered that the features of sample collection and handling, as well as the stages and duration of sample storage could vary widely in different biobanks. Among the omics approaches, genomics is the least sensitive to preanalytical procedures, while the integration of the different collections with dissimilar processing conditions in the transcriptomic, metabolomic, or microbiome studies can lead to false findings and erroneous conclusions. However, even in genomic studies, sub-optimal handling/storage conditions may lead to the decrease of quality of sequencing including sequencing failure, and increased C>A/G>T errors rate resulting in low-quality of reads. Parameters such as the type of blood vacuum tube preservative, the type of stored biomaterial, the storage conditions (especially temperature and duration of storage), and transportation may affect the quality of genomic studies. The availability of a detailed description of pre-analytical procedures allows us to avoid the aforementioned problems. Representation of this information in an encoded form, for example, using the Standard PREanalytical Code (SPREC) system [132], may provide additional advantages. Availability of detailed sample processing information may enhance efficient correction of batch effects during data integration.

Taken together, trans-biobank data integration seems extremely important for the validation of GWAS findings and the identification of robust genetic markers for common diseases and quantitative phenotypes. At the same time, the integration procedure has a few important caveats which have to be carefully considered during statistical analyses and interpretation of findings.

## 5. Conclusions

The empowerment of biobanks and the expansion of their collections with high-throughput data facilitates more extensive studies and implementation of their results in clinical practice. Population allele frequency statistics, genotype panels for imputation and linkage-based analyses, population-centered PRS, and other aforementioned results of biobank applications in genomics highlight the importance of biobanks in medical genetics, genomics, and healthcare.

We reviewed some successful biobank initiatives across the world and illustrated recent findings obtained using biobank data. Indeed, this review could not cover all results which were generated with the help of biobanks over the last years. Hence, we only discussed those publications that, in our opinion, provide the most significant and noteworthy examples. As indicated by this review, biobanks are more commonly used in the field of genome-wide association studies. However, recent progress in the field of functional genomics and metagenomics, such as the rapid spread and introduction of microbiome analysis in both research and clinical practice, may soon change the way biobank samples and data are analyzed and handled. Furthermore, the emergence of trans-biobank, cross-ancestry research projects indicate a general trend towards data integration and focus on robustness and reproducibility of results. Moreover, the enlargement of genomic studies by merging biobank data allowed researchers to investigate both common and population-specific genomic patterns of health and disease across various populations.

Our analysis was not mostly dedicated to internal issues of biobanking, such as sample collection, storage space, and capacity, sample and data management, and ways of communication with participants. These aspects, however, are no less important for the further full integration of biobanks and genomics into clinical practice. We believe that tight collaboration between geneticists, medical professionals, and policymakers will bring biobanking and, therefore, personalized precision medicine to our everyday lives.

## Figures and Tables

**Figure 1 jpm-12-02040-f001:**
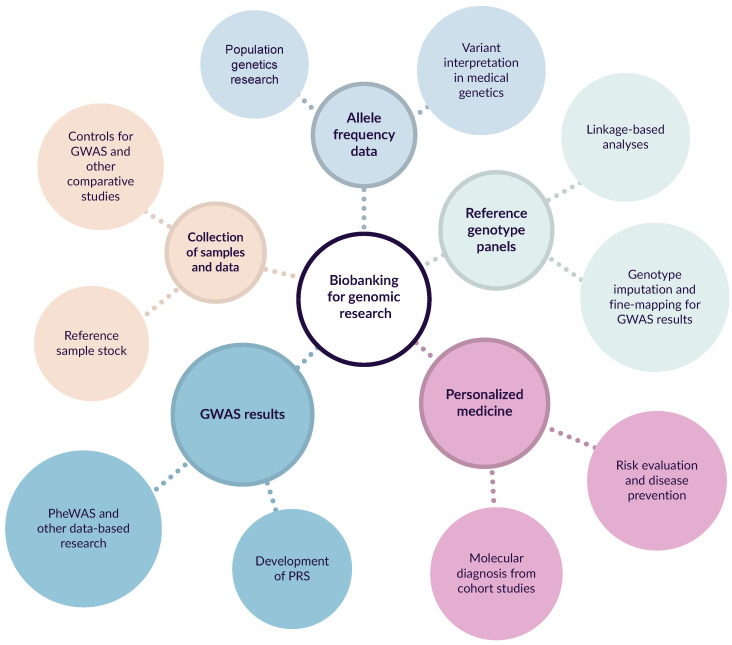
A diagram showing five major types of biobank applications for genomic research and medical genetics.

**Figure 2 jpm-12-02040-f002:**
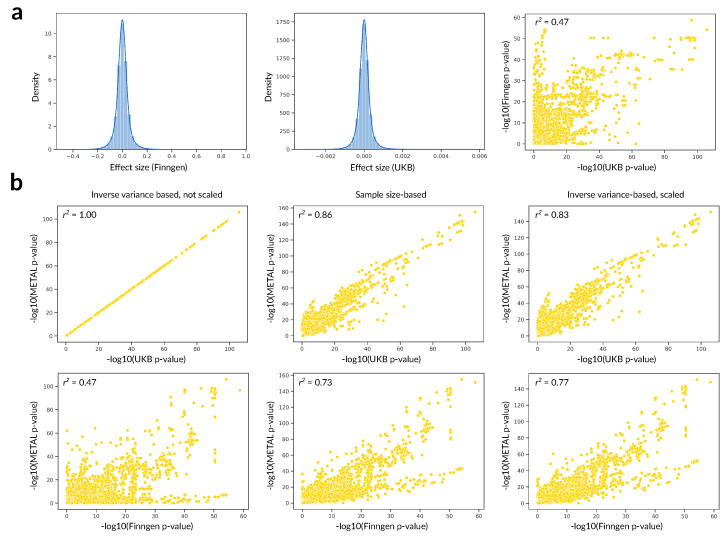
Bias in meta-analysis due to differences in data preprocessing. (**a**) Descriptive statistics of the raw data. Shown are (from left to right) effect size distribution in FinnGen, effect size distribution in UKB, and a scatterplot of *p*-values from UKB and FinnGen. (**b**) Scatterplot showing a comparison of METAL meta-analysis *p*-values for (from left to right) inverse variance method using raw data, sample size-based method, and inverse variance method using scaled data (scaling was performed by multiplying UKB effect size by the ratio of mean effect size from FinnGen to UKB). Note the extremely high degree of correlation of meta-analysis p-values and UKB *p*-values in case of inverse variance test and original UKB effects. Phenotypes used for analysis: other (seronegative) rheumatoid arthritis, wide (RHEUMA_OTHER_WIDE) from FinnGen, and other rheumatoid arthritis (M06) from UKB. Data and code used for the analysis presented in this figure are publicly available in the repository at https://github.com/TohaRhymes/meta-analysis-methods-comp, accessed on 27 November 2022.

**Table 1 jpm-12-02040-t001:** Candidate genetic variants associated with COVID-19 related quantitative traits in a cohort of Russian patients.

Biobank	Location	Number of Participants ^a^	Cohort	Biosamples	Sample Availability	Omics Data	Example Studies
UK Biobank	UK	500,000	closed, population aged 40–69	blood, urine, saliva	yes	genotyping array, WGS, WES, metabolomics, telomere length	Bycroft et al., 2018 [37]; Wells et al., 2019 [38]; Watanabe et al., 2019 [26]; van Hout et al., 2020 [39]; Shikov et al., 2020 [27]; de Vincentis et al., 2022 [40]; Halldorsson et al., 2022 [15]
BioBank Japan	Japan	267,307	closed, patient-based	serum, DNA, tumor tissues	yes	genotyping array, WGS, metabolome	Ishigaki et al., 2020 [41]; Matoba et al., 2020 [42]
FinnGen	Finland	538,600	open, general population, 15 disease-specific cohorts	depends on sample collector ^b^	depends on sample collector ^b^	genotyping array	Desch et al., 2020 [43]; Kurki et al., 2022 [44]; Sun et al., 2022 [45]
Estonian biobank	Estonia	200,000	open, adult population	whole blood and fractions, DNA, RNA	yes	WGS, WES, genotyping array, metabolomics (NMR), RNA seq., genome-wide methylation arrays, genome-wide gene expression array	Alver et al., 2019 [29]; Reisberg et al., 2019 [46]
China Kadoorie Biobank	China	512,891	closed, residents of 5 urban and 5 rural provinces aged 30–79	blood	no	genotyping array, WGS	Spracklen et al., 2020 [47]; Giannakopoulou et al., 2020 [48]; Zhu et al., 2021 [49]; Li et al., 2022 [50]
Tohoku Medical Megabank Project	Japan	157,000	closed, adult residents of Miyagi and Iwate Prefecture, three-generation cohort	blood fractions, urine, saliva, breast milk, dental plaque, stimulated T-cells, and EBV-transformed B-cells	yes	WGS, genotyping array, metabolomics (NMR, LC-MS), genome-wide methylation arrays	Watanabe et al., 2018 [51]; Tadaka et al., 2019 [52]; Tadaka et al., 2021 [53]; Kawame et al., 2022 [28]; Ohneda et al., 2022 [54]; Park, 2022 [55]
Taiwan Biobank	Taiwan	181,635	open, population aged 20–70, patient-based	blood, urine, saliva	yes	genotyping array, WGS, DNA methylation, HLA typing, metabolomics	Weiet al., 2021 [56]; Lee at al., 2022 [57]; Juang et al., 2021 [58]
LifeLines Cohort Study	Netherlands	167,000	closed, residents of the northern part of country	blood, urine, saliva, scalp hair	yes	genotyping array, WGS, microbiome data	Bonder et al., 2016 [59]; Imhann et al., 2016 [60]; Zhernakova et al., 2018 [61]
National Biobank of Korea	South Korea	1,051,787	population-based, patient-based	blood fractions, urine, saliva, DNA, tissue	yes	genotyping array	Namet al., 2022 [62]; Moon et al., 2019 [63]
Karolinska Biobank	Sweden	700,000	collection-specific	whole blood and fractions, urine, saliva, DNA	depends on collection	genotyping array	Bonfiglio et al., 2018 [64]
HUNT Biobank	Norway	120,000	open, adolescent and adult residents of Trøndelag	whole blood, plasma, serum, urine, saliva, feces, DNA	yes	genotyping array	Nielsen et al., 2018 [65]; Nielsen et al., 2020 [66]; Surakka et al., 2020 [67]
Canadian Partnership for Tomorrow’s Health	Canada	331,359	open, residents of 9 provinces aged 30–74	whole blood and fractions, urine, saliva, dry blood spots, nail fragments	yes	genotyping array	Lona-Durazo et al., 2021 [68]; Joseph et al., 2022 [69]
All of Us Research Program	USA	348,000	open, adult minority population	whole blood, urine, saliva	no	WGS, genotyping array	n.a.
BioVU	USA	275,000	open, pediatric and adult patient-based	DNA	yes	genotyping array	Zhenget al., 2021 [70]; Goldstein et al., 2020 [71]; Krebs et al., 2020 [72]
Penn Medicine BioBank	USA	52,853	open, adult patient-based	blood, tissue	yes	genotyping array, WES	Parket al., 2021 [73]; Akbari et al., 2022 [74]

## Data Availability

All data and code pertinent to the new analysis are available at https://github.com/TohaRhymes/meta-analysis-methods-comp, accessed on 27 November 2022.

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
