# Peer review of "Biobanking as a Tool for Genomic Research: From Allele Frequencies to Cross-Ancestry Association Studies"

_jpm, 2022, doi:10.3390/jpm12122040_

Round 1

Reviewer 1 Report

It is a well written paper. However, it is not clear how the biobanks described in the current paper were selected, plus the choice between describing separately and those that are briefly discussed in 3.8.

In addition, a number of topics that should be addressed in a paper on biobanking and genomic research are missing:

- Very little is said about return of results (line 99-100 / line 208-209 / line 274). Nothing is even mentioned about the policy of various biobanks with regard to unsolicited findings.

- In line 99-100 is stated '...returning genomic results to partcipants…' and in line 208-209 ‘…receive the results of the research…’. Does this apply to all results including variants of uncertain significance (VUS)?

- The article ends with a paragraph about what the paper didn't focus on, such as sample collection. When listing the various biobanks, however, you would expect information about sample collection and handling (~pre-analysis). This is also relevant when it comes to the concerns listed (line 469-504), but again no mention is made of the effect of differences between procedures for sample collection and handling when pooling biomaterial from different biobanks.

Author Response

Reviewer 1

It is a well written paper. 

Authors: We would like to thank the Reviewer for high assessment of our work and the valuable feedback.

However, it is not clear how the biobanks described in the current paper were selected, plus the choice between describing separately and those that are briefly discussed in 3.8.

Authors: In our work, we made an attempt to select and review the largest genomics-focused biobanks or multi-biobanks projects around the globe, particularly focusing on those involved in the Global Biobank Meta-analysis Initiative (GBMI). The largest (and arguably most cited ones) are discussed separately.. During revision, we expanded both the table and the text and included such prominent initiatives as Taiwan Biobank, National Biobank of Korea, BioVU, and Penn Medicine Biobank which are all part of GBMI. We also corrected the wording in the beginning of section 3 to define the selection criteria more clearly (lines 139-144).

In addition, a number of topics that should be addressed in a paper on biobanking and genomic research are missing:

- Very little is said about return of results (line 99-100 / line 208-209 / line 274). Nothing is even mentioned about the policy of various biobanks with regard to unsolicited findings.

- In line 99-100 is stated '...returning genomic results to partcipants…' and in line 208-209 ‘…receive the results of the research…’. Does this apply to all results including variants of uncertain significance (VUS)?

Authors: We have expanded the section describing the applications of biobanks to include more information on return of results, unsolicited findings, and the issue of variants of uncertain significance (lines 100-134)

- The article ends with a paragraph about what the paper didn't focus on, such as sample collection. When listing the various biobanks, however, you would expect information about sample collection and handling (~pre-analysis). This is also relevant when it comes to the concerns listed (line 469-504), but again no mention is made of the effect of differences between procedures for sample collection and handling when pooling biomaterial from different biobanks.

Authors: We have added a paragraph discussing the importance of the pre-analytical stage of sample processing (lines 590-604)

Reviewer 2 Report

The manuscript by Lazareva et al. describes the current situation of biobanking across the world as regards the use of all the wealth of information related to genetics in these health platforms. The authors stress that biobanks are poised to support a lot of advances in clinical practice and biomedical research and that trans-biobank projects are an important way forward.

The text is clear and well organized. I have just two comments:

(1) Use of HPO. The authors describe the use of Human Phenotype Ontology terms as "promising" with the caveat that those terms are biased towards Mendelian diseases and hardly used in many biobanks. From my point of view, it is highly advisable that HPO terms become widely used in the biobanking community to help leverage all the infromation in artificial-intelligence (AI)-assisted analyses. This will become the norm in the immediate future and biobanks that do not adopt HPO will be left out and their resources will not be used. I suggest that the authors mention this at the appropriate point (lines 486-488).

(2) Another interesting use overlooked by the authors is that clever use of genotyping informatioj in the biobank cohorts can be used to derive population haplotypes, which in turn help very much perform genomic analyses. This could also be discussed within section 2 (Applications)

Author Response

Reviewer 2

The manuscript by Lazareva et al. describes the current situation of biobanking across the world as regards the use of all the wealth of information related to genetics in these health platforms. The authors stress that biobanks are poised to support a lot of advances in clinical practice and biomedical research and that trans-biobank projects are an important way forward.

Authors: We thank the Reviewer for such a positive assessment of our work.

The text is clear and well organized. I have just two comments:

(1) Use of HPO. The authors describe the use of Human Phenotype Ontology terms as "promising" with the caveat that those terms are biased towards Mendelian diseases and hardly used in many biobanks. From my point of view, it is highly advisable that HPO terms become widely used in the biobanking community to help leverage all the infromation in artificial-intelligence (AI)-assisted analyses. This will become the norm in the immediate future and biobanks that do not adopt HPO will be left out and their resources will not be used. I suggest that the authors mention this at the appropriate point (lines 486-488).

Authors: The discussion of HPO terms has been expanded, and the necessity of standardization in the context of AI-driven research has been mentioned (lines 566-578).

(2) Another interesting use overlooked by the authors is that clever use of genotyping informatioj in the biobank cohorts can be used to derive population haplotypes, which in turn help very much perform genomic analyses. This could also be discussed within section 2 (Applications)

Authors: in fact, this application was briefly mentioned in the section 2 of the original manuscript (lines 64-70). We have revised this part of the text to include more details (lines 65-72 of the revised manuscript)

Round 2

Reviewer 1 Report

Thanks for the clarification and the adjustments made to the paper.